# Lettuce (*Lactuca sativa*) productivity influenced by microbial inocula under nitrogen-limited conditions in aquaponics

Jessica A. Day[1☯], Christian Diener[1☯], Anne E. Otwell[1☯], Kourtney E. Tams[2], Brad Bebout[3], Angela M. Detweiler[3,4], Michael D. Lee[3,5], Madeline T. Scott[6], Wilson Ta[6], Monica Ha[6], Shienna A. Carreon[6], Kenny Tong[6], Abdirizak A. Ali[6], Sean M. Gibbons[1,7,8]*, Nitin S. Baliga[1,9,10,11,12]*

1 Institute for Systems Biology, Seattle, Washington, United States of America, 2 St. Edward's University, Environmental Science and Policy, Behavioral and Social Sciences, Austin, Texas, United States of America, 3 National Aeronautics and Space Administration, Exobiology Branch, NASA Ames Research Center, Mountain View, California, United States of America, 4 Bay Area Environmental Research Institute, Moffett Field, California, United States of America, 5 Blue Marble Space Institute of Science, Seattle, Washington, United States of America, 6 Seattle Youth Employment Program, Seattle, Washington, United States of America, 7 eScience Institute, University of Washington, Seattle, Washington, United States of America, 8 Department of Bioengineering, University of Washington, Seattle, Washington, United States of America, 9 Department of Microbiology, University of Washington, Seattle, Washington, United States of America, 10 Department of Biology, University of Washington, Seattle, Washington, United States of America, 11 Molecular Engineering and Sciences, University of Washington, Seattle, Washington, United States of America, 12 Lawrence Berkeley National Laboratories, Berkeley, California, United States of America

☯ These authors contributed equally to this work.
* sgibbons@isbscience.org (SMG); nbaliga@isbscience.org (NSB)

**Data Availability Statement:** R notebooks and scripts to reproduce the analysis starting from the raw FASTQ files are provided at https://github.com/gibbons-lab/aquaponics. The analysis of the validation samples can be found in the same

## Abstract

The demand for food will outpace productivity of conventional agriculture due to projected growth of the human population, concomitant with shrinkage of arable land, increasing scarcity of freshwater, and a rapidly changing climate. While aquaponics has potential to sustainably supplement food production with minimal environmental impact, there is a need to better characterize the complex interplay between the various components (fish, plant, microbiome) of these systems to optimize scale up and productivity. Here, we investigated how the commonly-implemented practice of continued microbial community transfer from pre-existing systems might promote or impede productivity of aquaponics. Specifically, we monitored plant growth phenotypes, water chemistry, and microbiome composition of rhizospheres, biofilters, and fish feces over 61-days of lettuce (*Lactuca sativa var. crispa*) growth in nitrogen-limited aquaponic systems inoculated with bacteria that were either commercially sourced or originating from a pre-existing aquaponic system. Lettuce above- and below-ground growth were significantly reduced across replicates treated with a pre-existing aquaponic system inoculum when compared to replicates treated with a commercial inoculum. Reduced productivity was associated with enrichment in specific bacterial genera in plant roots, including *Pseudomonas*, following inoculum transfer from pre-existing systems. Increased productivity was associated with enrichment of nitrogen-fixing *Rahnella* in roots of plants treated with the commercial inoculum. Thus, we show that inoculation from a

repository. More complex algorithms such as the Expectation-Maximization algorithm are available in a dedicated R package along with documentation at https://github.com/gibbons-lab/mbtools. Raw sequencing data (study and validation data) has been deposited in NCBI Sequence Read Archive (SRA) under the project ID PRJNA641448.

**Funding:** This research was possible due to the support and resources provided by Ray Williams and the Black Farmers Collective, Jeff King and the Microsoft Giving Campaign, Fred Hutchinson Cancer Research Center, Seattle Youth Employment Program, CrowdRise donors, the National Science Foundation (NSF MSB-1237267, MCB-1616955, MCB-1518261, DB-1262637, DB-1565166, MCB-1330912), a Washington Research Foundation Distinguished Investigator Award (supporting CD and SMG), and a Scientific Innovation Fund grant from the NASA Office of the Chief Scientist to Brad M. Bebout. The funders had no role in study design, data collection and analysis, decision to publish, or preparation of the manuscript.

**Competing interests:** The authors have declared that no competing interests exist.

pre-existing system, rather than from a commercial inoculum, is associated with lower yields. Further work will be necessary to test the putative mechanisms involved.

## Introduction

Sustainable food production has been on the rise in recent decades as traditional agricultural practices, which contribute to large-scale environmental degradation and enormous resource consumption, fall short of fulfilling the demands of our growing human population [1]. Aquaponics offers a sustainable alternative to traditional food production methods by combining hydroponic plant cultivation with aquaculture in a semi closed-loop system [2] that minimizes water and fertilizer use, increases agricultural efficiency [3], and does not require arable land. Central to the health of fish and plants in these systems are microorganisms, which drive many critical functions such as nitrogen cycling, plant growth promotion, disease resistance, and nutrient uptake; however, a deeper understanding of microbial community composition and function in aquatic agricultural systems is central to engineering and scaling-up efficient, sustainable food systems with low natural resource dependence [4]. While some work has been conducted on microbial communities in hydroponics [5] and aquaponics [4, 6], our knowledge of the microbial ecology of aquaponics is mainly grounded in soil-based agricultural research [7–9].

Interest among researchers and growers in aquaponic microbes has been focused on initiating nitrogen cycling and promoting plant growth via inoculation with plant growth promoting microbes (PGPMs). For this reason, one of two inoculation strategies are traditionally used to initiate cycling: 1) addition of commercially-derived nitrifying bacteria (*Nitrosomonas*, *Nitrobacter*, and *Nitrospira*) or 2) transfer of established bacteria from existing, healthy aquaponic systems. Despite the inclusion of PGPMs, a 2018 international survey found that 84.4% of aquaponic growers observed disease in their systems, of which 78.1% could not identify the causal agent [10]. Therefore, understanding the effect that microbial transfer has on production in aquaponics is crucial not only to establish best practices and increase commercial profitability by way of improving efficiency, but also to decrease loss due to disease. Of the growers who observed disease, a mere 6.2% used pesticides or biopesticides against plant pathogens and relied, instead, on non-curative actions, likely due to a lack of knowledge among aquaponic growers regarding disease control measures [10]. Knowledge of key associations between microbial genera and productivity throughout early stages of system establishment could enable the development of diagnostic tools for monitoring microbiome composition, potentially aiding in early detection and prevention of system collapse.

Institute for Systems Biology (ISB) established Project Feed 1010 (PF1010) to promote education and research around sustainable food systems, such as aquaponics, to help combat global food insecurity. Through ISB high school internships supported by the Seattle Youth Employment Program, small-scale aquaponic experiments were designed and carried out in collaboration with researchers at the National Aeronautics and Space Administration (NASA) Ames Research Center to test how two inoculation strategies impacted productivity. The Institutional Animal Care and Use Committee (IACUC) protocol limited the number of fish per system to prioritize animal welfare, which meant that our aquaponic systems were nitrogen-limited compared to commercial systems. We examined how microbiome transfer from either pre-existing systems or commercial inoculum promoted or impeded plant productivity. We compared lettuce (*Lactuca sativa var. crispa*) production in these two distinct systems—those

inoculated with the biofilter media from an established, fully cycled aquaponic system ("established inoculum treatment" or "EIT") and those inoculated with a commercially-available microbial consortium ("commercial inoculum treatment" or "CIT").

## Results and discussion

An overview of the aquaponics systems and experimental design is shown in Fig 1A. Over the 61-day study period, lettuce growth (height, number of leaves, and root length) was significantly reduced in all EIT replicates compared to growth in CIT replicates (Fig 1B–1D). Physicochemical properties did not significantly differ across aquaponic systems over the study period (S1 Fig, all individual Welch t-test p>0.07), making it unlikely that the observed growth disparity can be explained by system-wide biogeochemical parameters. Therefore, we explored potential associations between microbial community composition and plant growth parameters.

We first investigated whether microbiome transfer affected the nitrogen transformation process. As anticipated, all systems showed low levels of nitrogen due to the limited number of fish allowed per system to maintain compliance with IACUC. One CIT system (tank 3) showed a transient nitrogen spike associated with the death of one fish (Fig 2A). However, areas under the nitrate curves (AUNC) were not significantly different between EIT and CIT tanks (Welch t-test p = 0.3) and AUNC variance was explained slightly better by the number of dead fish in each tank ($R^2$ of 0.37 vs 0.23, see Materials and methods). Similarly, there was

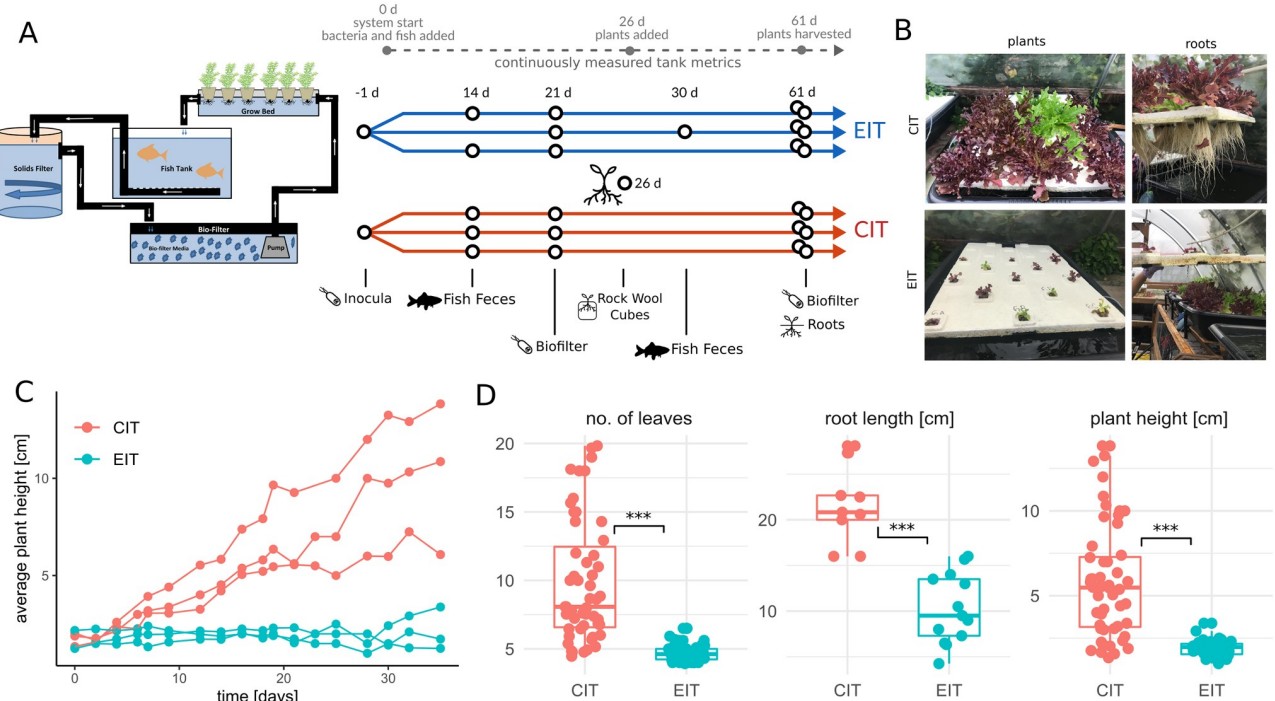

**Fig 1. Aquaponic system design and plant phenotypes.** (A) Aquaponics sampling timeline and system design. Left shows aquaponics system design and right sampling strategy. Gray circles on top denote experimental events and every black empty circle denotes a single sample. One fish feces sample could not be extracted at day 14 and was resampled at day 30. (B) Representative images of *L. sativa* plants and roots after a month of growth in systems with different inocula. (C) Plant growth over time. Each dot denotes the average plant height for a single aquaponic system taken at the indicated time point. Measures from the same tank are connected by lines. Gray line denotes growth in the prior aquaponics system that was used as the source for EIT. (D) Plant growth measures by inoculum. Each point denotes an average value measured in a single tank at a single time point (n = 104, 22, 105 for leaves, root length, and plant height, respectively). Stars denote significance under a Mann-Whitney U test (all p<0.001).

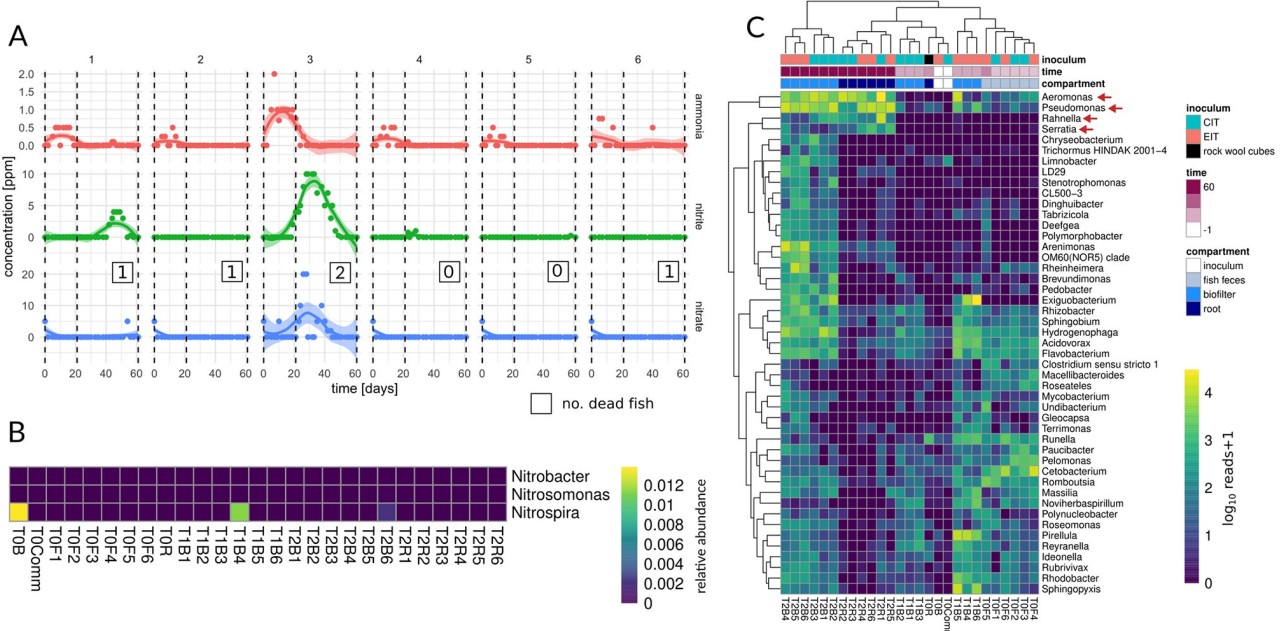

**Fig 2. Nitrogen cycling and microbial composition.** (A) Shown are measurements for ammonia, nitrite and nitrate throughout the experiment. Colored lines denote smooth fit from a LOESS regression and filled areas denote 95% confidence intervals of the regression. Numbers of the panels denote tanks. Tanks 1–3 are CIT whereas tanks 4–6 are EIT. Dashed lines denote the three time points used for microbiome sampling. Boxed numbers denote the number of fish that died in the tank. (B) Relative abundance (fraction of total reads per sample) of known nitrifying taxa. (C) Abundances of ubiquitous bacterial genera across the samples (present in at least 2 samples at an abundance >300 reads). Colors of cells denote the normalized abundance on a base 10 log-scale. Sample names are composed of sampling group ID (e.g. T1), compartment (B = biofilter, F = fish feces, R = root, Comm = commercial inoculum) and tank number (1–6). Orange arrows denote genera of interest. Black fill color denotes initial root sample from rock wool cubes (not part of inoculation strategy).

no significant difference in the number of fish that died between CIT and EIT systems. Previous studies of established aquaponic systems found a ubiquitous presence of nitrifiers such as *Nitrosomonas*, *Nitrobacter*, and *Nitrospira*, albeit in low abundances, and these organisms are described as major drivers of plant growth [6, 11]. We only detected nitrifying taxa in the established inoculum itself, as well as in two EIT samples from days 21 and 61, respectively (Fig 2B). Instead, biofilter samples were dominated by Proteo- and Cyanobacteria (S2 Fig and Fig 2C). Even though our protocol was validated to detect nitrifiers (S3 Fig), we also found no nitrifiers in the commercial inoculum, which was dominated by *Rhodanobacter*, a genus containing known denitrifiers [12]. This suggests that the improved plant growth in CIT systems was likely independent of nitrifying bacteria. We also observed low nitrogen levels in our end-point plant nutrient analyses (S2 Table) and water chemistry (S1 Fig), suggesting that nitrogen was limiting. This finding could explain the increased abundance of nitrogen fixers, such as *Rahnella*, and reduced abundance of nitrifying species (Fig 2B and 2C). We hypothesize that in nitrogen-limited aquaponic systems, nitrogen fixing bacteria may play an important growth-promoting role in supplementing the limited ammonia produced by fish by fixing atmospheric nitrogen and producing additional ammonia, which is a well-known nitrogen source for *L. sativa* [13].

In examining whether microbiome transfer affected establishment of microbial communities in new systems, we found alpha-diversity increased with time in all compartments and achieved similar values for CIT and EIT tanks in biofilters and roots at day 61 (Fig 3A). This temporal development of alpha-diversity was not an artefact of a bias due to sequencing depth

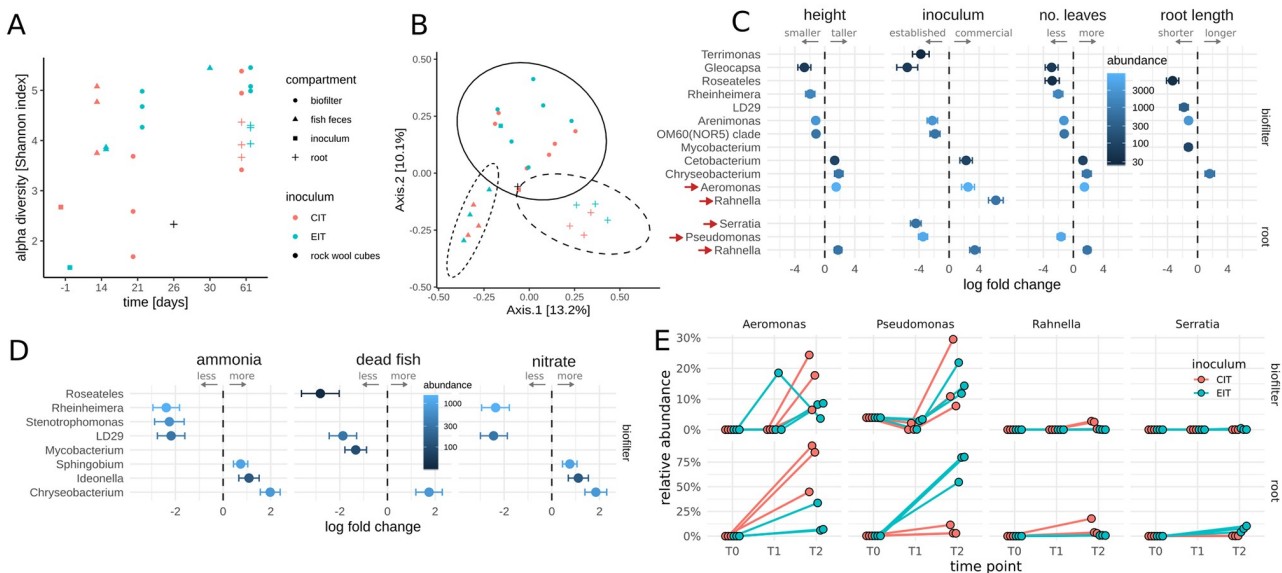

**Fig 3. Amplicon sequencing of the full-length 16S rRNA gene across the aquaponic systems.** (A) Alpha-diversity (Shannon index) over time. Colors and shapes denote initial microbial sources and sampling compartments respectively. All samples were rarefied to 3,000 reads each. (B) (B) PCoA plot of individual samples from all time points. Colors and shapes are the same as in panel A. All samples were rarefied to 3,000 reads each. Ellipses denote 95% confidence interval from Student t-distribution separating compartments. (C) Significant associations (FDR adjusted p<0.05) between bacterial genera and plant growth metrics or inoculum. Points denote the association coefficient in the respective regression and error bars denote the standard error of the coefficient. Fill color denotes average abundance across all samples. (D) Significant associations (FDR adjusted p<0.05) between bacterial genera and measures associated with nitrification. Circles denote the association coefficient in the respective regression and error bars denote the standard error of the coefficient. Fill color denotes average abundance across all samples. (E) Time course of selected genera associated with plant growth and inoculum. Time point zero is shared between all samples and denotes initial inoculum for biofilter and initial plant microbial composition for roots. Only genera with more than 300 reads in at least 2 samples were considered in A-E (see Materials and methods).

(S4 Fig). Aquaponic compartments each had distinct microbial compositions (Fig 3B). A total of 44% of variation in beta-diversity was explained by a combination of compartment (25%), inoculum (11%), and an interaction term of the two (8%; all PERMANOVA p values < 0.02). Conversely, prior studies in nitrogen-replete systems found that the microbial composition of different compartments in established systems, with the exception of fish feces, were quite similar [6].

Given the differences in microbial composition between EIT and CIT systems, we hypothesized there may be a negative effect of microbial transfer from a prior system on plant growth, which acts independent of nitrogen concentration and cycling timeline. Thus, we investigated whether the difference in plant growth could be explained by an enrichment in potentially growth-inhibiting bacteria from the established system inoculum. This microbiome-inhibition hypothesis is similar to negative plant-soil feedback (NPSF), which has long been studied in both field and agricultural soils [14–17]. NPSF is thought to be characterized by enrichment of species-specific plant pathogens or the accumulation of allelochemicals that limit plant productivity in successive generations grown in the same soil [14–17]. This self-inhibitory process promotes plant community diversity by allowing sub-dominant species to thrive [14, 15], and has served as a justification for crop rotations as a practice in soil-based agriculture [18]. Thus, we hypothesize that a similar negative feedback phenomenon may be responsible for the reduced aquaponics productivity.

In order to explore putative biotic mechanisms for the growth-promotion and negative-feedback hypotheses outlined above, we examined whether there were distinct bacterial genera in the plant roots that were associated with inoculum source and plant growth phenotypes. To

that end we conducted negative binomial likelihood-ratio tests (LRT) between plant pheno-types and microbial abundances with DESeq2 [19]. Association tests between bacterial genus-level abundances and plant growth measures revealed that three highly abundant genera in the roots, namely *Pseudomonas*, *Rahnella* and *Serratia*, were significantly associated (LRT FDR corrected p<0.05) with plant growth (Fig 3C, see Materials and methods). *Pseudomonas* and *Serratia* relative abundances were higher in the EIT systems and were thus associated with diminished plant growth, whereas *Rahnella* was more abundant in CIT systems and was there-fore associated with improved plant growth (Fig 3C). Some isolated incidents of fish death were observed across all systems, though it is unclear if this led to altered nitrate abundances (area under the nitrate curve, Spearman rho = 0.73, p = 0.1). However, no bacterial abun-dances in the roots were correlated with fish mortality, ammonia, or nitrate levels (all FDR cor-rected p>0.1), suggesting that establishment of the rhizosphere microbiota was independent of fish mortality or system-wide nitrogen cycling (Fig 3D). Although *Pseudomonas*, *Rahnella* and *Serratia* were all associated with plant growth metrics, only members of the genus *Pseudomonas* were 1) previously described as plant pathogens [20–22] and 2) found in both initial inoculum types. Across time points, we observed a relatively uniform accumulation of *Pseudomonas* in biofilters, with all biofilter microbiomes consisting of 5–30% *Pseudomonas* (Fig 3E). However, detection of *Pseudomonas* in the rhizosphere was highly dependent on inoculum type. EIT rhizosphere microbiomes were dominated by *Pseudomonas* (with 50–80% *Pseudomonas*, LRT FDR-corrected p<0.05), whereas CIT rhizosphere microbiomes contained less than 20% *Pseudomonas*. Instead, CIT rhizospheres were enriched for *Aeromonas* and nitro-gen-fixing *Rahnella*, which were associated with improved plant growth (Fig 3E).

In summary, we saw two divergent but not necessarily mutually exclusive signals associated with inoculum type and aquaponic productivity. First, the genus *Pseudomonas* was enriched within biofilters across all systems, but only dominated the *L. sativa* rhizosphere in systems inoculated with microbes from an established aquaponic system, suggesting that EIC-specific growth-inhibitory pseudomonads may be responsible for a reduced yield in EIC-treated sys-tems. However, our MinION sequencing data cannot resolve taxonomy beyond the genus level, so we are unable to verify whether or not specific bacterial pathogens were present in our systems. Furthermore, it is possible that viral, fungal, or other eukaryotic plant pathogens could be enriched in the established inoculum, as we did not collect data on non-bacterial microbiota in this study. Second, we saw enrichment of nitrogen-fixing *Rahnella* in CIT rhizo-spheres. *Rahnella* species are known to form associations with plant roots, promote plant growth, and have been found with other diazotrophs in nitrogen-fixing root nodules [23, 24]. Given the low levels of nitrate and ammonia in our aquaponic systems, *Rahnella* strains may have promoted *L. sativa* growth.

Overall, further work is necessary to distinguish whether associations between aquaponics productivity and inoculum treatment are due to bacterial plant pathogens, growth-promoting microbes, or factors including other plant pathogens (e.g., viral, fungal, protist) or archaeal species (such as ammonia-oxidizing archaea), which were not quantified in our study. Addi-tional studies with larger sample sizes, multi-omic data collection, and bacterial isolate charac-terization will be necessary to identify pathogenic or growth-promoting strains, establish causality, and test our mechanistic hypotheses. However, our findings indicate that CIT aqua-ponics systems showed greater yields when compared to EIT systems under nitrogen-limiting conditions and therefore that established inocula are not optimal for nitrogen-limited aquapo-nic systems. The decreased yield (an approximate 36-fold decrease in plant biomass; S1 Table) observed in this study could be financially devastating to aquaponic farmers [25]. While most commercial aquaponic systems are run under higher nitrogen levels than in our study, it is likely that other research institutions will follow similar animal welfare standards in future

aquaponic studies and therefore nitrogen will be limited. Therefore, our work is an important step towards characterizing aquaponic systems operated for research purposes, where more controlled experiments can be performed compared to commercial systems. Moving forward, metagenomic analyses, metabolomics, strain isolation, and whole-genome sequencing will provide deeper insights into the specific strains, functional genes, and small molecules involved in both positive and negative plant-inoculum feedbacks [26] and could lead to targeted strategies for engineering the microbial ecology of aquaponics systems to improve yield and resource use efficiency.

## Materials and methods

### Aquaponic system design

Six independent aquaponics systems were constructed using a 4-compartment design (Fig 1A), with three replicate systems treated with a commercial inoculum and three with an established inoculum, in an open-air greenhouse in Seattle, Washington. These treatments will henceforth be labeled CIT (commercial inoculum treatment) and EIT (established inoculum treatment). Each system housed one aquaculture tank connected to one hydroponic unit utilizing the deep water culture method. These units were separated by a biofilter, which housed approximately 60 L of K3 filter media, as well as an 18 L solids filter. The experiment commenced when two adult Red Nile Tilapia (*Oreochromis niloticus*) were added to each system. On the same day, the 6 systems were divided into two sets of triplicate systems and inoculated with either Microbe-Lift commercial bacteria (tanks 1–3; "CIT") or established microbes from an existing aquaponic system (tanks 4–6; "EIT"). Four weeks later, 13 lettuce seedlings, which had been sprouted in rockwool cubes outside the systems 3 weeks earlier, were added to the floating trays (Beaver Plastics 28-hole 2ft x 4ft Lettuce Raft) in each system and allowed to grow 35 days.

### Animal care

Prior on-site research conducted at the Institute for Systems Biology under IACUC protocol number NB.01a.16 ("Educational Aquaponics"; OLAW #A4355-01 and AAALAC #001363) provided guidelines that were closely followed in this off-site study. Water chemistry, space, and fish food were all determined and maintained to optimize living conditions for all Nile Tilapia (*Oreochromis niloticus*) used in this study, and fish health and behavior were observed and recorded regularly. Fish were fed Stage 3 Intermediate AquaNourish Omnivorous Fish Feed pellets (Star Milling Co., Perris, California, USA; 37% crude protein, 10% crude fat, 2.2% crude fiber, 11% ash) every other day in an increasing amount (between 10–40 pellets per fish) during nitrogen cycling, so as not to produce excess, toxic ammonia at initial stages of cycling. Each approximately 203 L system housed two fish with a total length of 44 inches. To maintain compliance with our IACUC protocol, which prioritizes animal welfare, only two fish per system were used (the sum of their lengths being approximately 44cm). Because this stocking density is lower than that of most commercial aquaponic systems, this experiment was run under relatively nitrogen-limited conditions. Throughout this study, 5 fish died unexpectedly due to unknown causes in tanks 1, 2, 3, and 6. Water chemistry was ruled out as a cause, as there were no associations between water chemistry fluctuations and subsequent fish deaths. One possible cause may be stress-induced infection by a fish pathogen such as *Saprolegnia* or *Aphanomyces*. Although 16S rRNA analysis is not capable of detecting these fungal pathogens, *Pseudomonas* and *Aeromonas* were increasingly enriched throughout the study. Both of these genera are known to include species which can mitigate oomycete diseases in aquaculture by

acting as an antipathogen agent [27]. Fish deaths were included as factors in our statistical analyses, to ensure that our reported results were independent of these adverse events.

## Inoculation of systems

Systems in the CIT were inoculated with Microbe-lift (Cape Coral, Florida, USA) commercial bacteria, marketed as containing *Nitrosomonas*, *Nitrobacter*, and *Nitrospira*, whereas EITs were inoculated with biofilter media from a previously established, highly-efficient *L. sativa*-producing aquaponic system where nitrogen had been fully cycled for approximately 2 years. To inoculate these systems, 10 bioballs (approximately 105.8 ng/mL bacteria DNA) from the existing aquaponic system were collected and transferred to the biofilter of each new EIT system. On the same day, 30 mL of commercial bacteria (company recommendation) was added to the CIT biofilters.

## Water chemistry maintenance

Throughout the study period, water chemistry (pH, temperature, ammonia, nitrite, nitrate, chlorine, hardness, and alkalinity) and plant growth metrics (number of leaves, plant height, root length, and presence of disease or discoloration) were collected 3–4 times per week (S1 Fig). Water chemistry data was collected using test strips (Tetra EasyStrips 6-in-1 Aquarium Test Strips and Tetra EasyStrips Ammonia Aquarium Test Strips) and a multi-parameter probe (Hanna Instruments® GroLine Waterproof Portable pH/EC/TDS Meter). To account for evaporation and plant use, an average of 126.5 L of aerated, dechlorinated tap water was added to each tank throughout the experimental period. Approximately 45g of Instant Ocean® Sea Salt (Blacksburg, VA) was added to each tank 5 days into the experiment to reach a conductivity of approximately 900 uS/cm.

## Lettuce phenotypes

Height, number of leaves, discoloration, and pest pressure, were recorded for each plant every other day at a consistent time. Height was measured from the base of each stem to the natural crest of the tallest leaf. Leaf count included cotyledons and emerging buds. Root length was recorded at the midpoint and end of the experiment and was measured from the base of each rockwool cube to the longest root of each plant.

## Sample collection

Microbiome samples were strategically collected throughout the 61 day study period and analyzed from 3 compartments of each system: 1) plant roots in the grow bed, where root-associated bacteria are located, 2) biofilter media in the biofilter where bacterial nitrifiers typically carry out nitrification, and 3) fish feces from the solids filter, where gut-associated bacteria can be found (Fig 1A). Sampling time points were reflective of key chemical species transformations occurring during nitrification, and therefore supposed microbial community shifts, across the study period (pre-cycling, during cycling, and post-cycling; Fig 1A). The experiment ended 61 days after microbial inoculation and plants were harvested. At the end of the experiment, wet and dry mass (biomass) of plants were collected (S1 Table). Nutrient analysis of dried leaf samples was performed by the University of Missouri Soil and Plant Testing Lab (S2 Table).

## Sample processing

**Roots.**   Time point 0 for the roots was taken from rockwool cubes that seedlings were growing in prior to transfer to the aquaponic systems. Nine rockwool cubes were collected from the grow tray and all liquid was squeezed from the rockwool into a 50 mL falcon tube. Tubes were centrifuged at 4,000 xg for 10 min, supernatant was discarded, and pellets were flash frozen in liquid nitrogen and stored at -80C until DNA extraction. The final time point for the roots was taken at the time of system disassembly (and plant harvest). Roots were clipped at the base of all plants in each system and collected in 50 mL falcon tubes. PBS was added to the tube and samples were sonicated at low frequency (intensity 1) for 5 minutes total (five 30 second bursts, each followed by a 30 second rest period) in order to remove cells from roots. Following sonication, roots were removed from the tube and samples were centrifuged at 4,000 xg for 10 min, supernatant was removed, and pellets were flash frozen in liquid nitrogen and stored at -80C until DNA extraction.

**Fish feces.**   Fish feces were collected from the bottom of the aquaponic solids filter using a 25 mL serological pipette. Samples were centrifuged at 4,000 xg for 10 minutes to pellet the feces samples, supernatant was discarded, and pellets were flash frozen in liquid nitrogen and stored at -80C until DNA extraction.

**Biofilter.**   Ten bioballs were collected from the established aquaponics biofilter using sterile forceps and stored in PBS. A single bioball was vortexed at maximum speed for 2 minutes in a 50 mL falcon tube and then sonicated at low frequency (intensity 1) for 5 minutes total (five 30 second bursts, each followed by a 30 second rest period). Following sonication, the bioball was removed from the tube and another bioball was added in, vortexed, and sonicated. This process was repeated until cells from 10 bioballs were collected for each sample. Cells from the 10 bioballs were then collected as a single pellet through centrifugation at 4,000 xg for 10 minutes, supernatant was discarded, and pellets were flash frozen in liquid nitrogen and stored at -80C until DNA extraction.

## Analysis of nitrogen cycling

Continuous nitrogen measures for each tank were reduced to areas under the curve (AUCs) using the trapezoidal method. To identify the dominant covariates associated with nitrogen cycling, nitrate AUCs were regressed against a binary inoculum covariate and the number of dead fish with a linear model in R (formula "nitrate ~ inoculum + dead_fish"). Explained variance of each term was obtained from an ANOVA analysis on the fitted model.

## DNA extraction, 16S rRNA sequence analysis

Microbial genomic DNA was isolated from samples using two similar extraction kits marketed specifically for bacterial DNA isolation from agricultural environments (Samples T0F1, T0F2, T0F3, T0F4, T0F6, and T0B with Qiagen PowerSoil kit and all others with PowerBiofilm kit). 16S rRNA genes were amplified using universal primers suggested by MinION (27F 5′-AGAGTTTGATCMTGGCTCAG and 1492R 5′-CGGTTACCTTGTTACGACTT) [28]. A MinION Nanopore Sequencer was used for sequence analysis.

## Sequencing

Amplicons were aliquoted to a starting concentration of 1 ug and were further processed and sequenced according to ONT's 1D Native barcoding genomic DNA (with EXP-NBD103 and SQK-LSK108) protocol (v. NBE_9006_v103_revN_21Dec2016). Processing started with an

AMPure XP bead purification step and proceeded to end-repair/dA-tailing, barcoding ligation, and adapter ligation steps.

## Analysis

Basecalling for the raw MinION files was performed by Albacore (v2.0.2) to yield the corresponding FASTQ files. Reads were processed using "filterAndTrim" methods from DADA2 [29]. The first 10 bp of the 5' end were trimmed from all reads as they generally showed lower qualities. Raw reads were also trimmed to a maximum length of 1.5 kbp (the expected length of the 16S gene). Reads with more than 200 expected errors under Illumina error model (based on [30]) or more than 2 ambiguous base calls ("N" bases) were removed from the analysis (~70% of all reads passed these filters). PCR chimeras were removed using yacrd version 0.5.1 [31]. However, yacrd removed very few sequences (<1%) because the prior length cutoff likely removed the majority of PCR chimeras. The filtered reads were then aligned to the complete SILVA 16S database using minimap2 with the Oxford Nanopore preset allowing up to 100 alternative alignments per read [32, 33].

## Validation and mapping improvement for high error rates

We observed acceptable read qualities with a median of 15 corresponding to an average error rate of 3% under the Illumina model. However, quality measures are based on a predictive model and the nominal error rate is usually above 10% [34]. In order to evaluate the accuracy of nanopore sequencing on the full length 16S gene we used a small validation data set consisting of two biological replicates of a long-running established aquaponic system sequenced with the same nanopore protocol as well as V4-V5 Illumina amplicon sequencing. V4-V5 sequencing data (515F 5'–GTGYCAGCMGCCGCGGTAA, 926R 5'–CCGYCAATTYMTTTRAG TTT; [35]) were analyzed using the DADA2 pipeline and used as the ground truth. Due to the high error rate of nanopore sequencing we expected a lot of spurious assignments when mapping to a 16S reference database such as SILVA. In order to limit those spurious matches, we employed an Expectation-Maximization (EM) algorithm similar to what is used in kallisto but without correction for gene length due to the fixed length of the 16S gene [36] (implementation available at https://github.com/Gibbons-Lab/mbtools). Applying the EM algorithm led to a reduction of unique mappings for low abundance cutoffs compared to a "naive" counting algorithm that just uses the highest scoring match (S3A Fig). We observed good agreement of estimated abundances between Illumina and Nanopore sequencing down to the genus level if the taxon was observed in both sequencing technologies (S3B Fig, $R^2$ between 0.5–0.8, Spearman rho between 0.73–0.85, all ANOVA $p < 10^{-6}$). However, we also observed many spurious mappings only present in the nanopore data, with generally low abundances (S3B and S3C Fig). We found that using an abundance cutoff of read 300 counts removed >96% of all spurious mappings across all taxa ranks and >97% of all spurious mappings on the genus rank. This cutoff did not depend on library size, which itself varied between ~9.8K to ~125K reads per sample. Thus we employed this abundance cutoff throughout the manuscript. The final abundances as estimated by the EM algorithm were then annotated by the SILVA taxonomy down to the strain level where available and collapsed on the genus level [37]. We also verified whether the nanopore 16S primers were capable of identifying the same nitrifiers that were amplified by the V4-V5 primers used in the Illumina run. We found that all known nitrifying taxa identified by Illumina sequencing were also found in the nanopore data (S3D Fig).

## Diversity metrics

Diversity estimates often depend on the library size (sequencing depth) of the sample. Library sizes ranged from ~9.8K reads to >125K reads across samples. In order to rule out bias based on library size, we applied rarefaction to 9000 reads to all samples. We also used rarefaction curves to judge whether sampling depth was sufficient to saturate the alpha diversity measure, which we found to be the case for all samples (see plateau in S4 Fig).

## Association testing

Read abundances across samples were normalized using the DESeq2 "poscounts" normalization strategy (similar to a centered log ratio transformation) [19]. Association tests were performed using DESeq2 with a prior filtering step to remove bacterial genera with either very low average abundances or low prevalence across samples (mean abundance >10 across all samples and present in at least 2 samples). This more permissive filter was used to allow for effective shrinkage for DESeq2. Significance was judged by a negative binomial log ratio test (LRT) comparing the full model to a model lacking the association term. Finally, we only considered significant tests for those genera which showed abundances larger than the default cutoff of 300 reads in at least two samples. Association tests between bacterial genus-level abundances and response variables were performed for each of the 3 environments (biofilters, roots, fish feces). The tested response variables were 3 plant growth measures (height, root length, number of leaves), inoculum type (CIT, EIT), and putative confounders (number of dead fish, area under ammonia and nitrate curves). False discovery rate was controlled by the Benjamini-Hochberg method [38]. No associations in the fish feces passed an FDR cutoff of 0.1.

## Supporting information

**S1 Fig. Water chemistry, environmental parameters, and system inputs measured throughout the study period.** Lines denote LOESS smoothed curves for each inoculum and bands denote 95% confidence intervals of the regression. Indicated p-values were obtained from individual t-tests of CIT vs EIT systems.
(PNG)

**S2 Fig. Relative abundances for the 12 most abundant phyla in the data set.** Sample names are composed of sample group ID (e.g. T1), compartment (B = biofilter, F = fish feces, R = root, Comm = commercial inoculum) and tank number (1–6). Panels denote inoculum as used in Fig 1 of the main text.
(PNG)

**S3 Fig. Validation of nanopore sequencing on a set of aquaponics biofilter samples that were sequenced using Illumina and nanopore technologies (2 replicates each).** (A) In low abundances, the Expectation-Maximization (EM) algorithm identifies fewer unique references than the "naive" strategy of just selecting the highest scoring read. Each dot denotes the number of unique 16S sequences in the SILVA database that pass the abundance cutoff. (B) Abundances across sequencing protocols. Each dot denotes the abundance of a single taxon at the indicated taxonomic rank. Blue lines denote a linear regression for the taxa found in both sequencing technologies and gray errors denote the 95% confidence interval of the regression. The red boxplot summarizes the distribution of spurious mappings in the nanopore data (absent in the Illumina data). (C) Distribution of false positive mappings (mappings not observed in the Illumina data) in nanopore sequencing. The dashed red line denotes the used abundance cutoff that removed >96% of those spurious mappings. (D) Abundances of

nitrifying taxa in both sequencing protocols. Dots denote the sum in the two replicates. Abundances smaller than one denote taxa not detected in Illumina sequencing.
(PNG)

**S4 Fig. Rarefaction curves for all samples.** Points denote alpha diversity estimate (Shannon index) after subsampling to the specified depth. Lines denote LOESS smoothing regression lines for each individual sample. Colors and panels denote sampling time point and aquaponics system compartment, respectively. Endpoint of each curve denotes the actual depth of each sample.
(PNG)

**S1 Table. Final wet mass and dry mass (biomass) of *L. sativa* leaves in all replicates.**
(XLSX)

**S2 Table. Amount of 9 key nutrients found in *L. sativa* leaves at the end of the 61-day study period.** *Low concentrations compared to *L. sativa* soil-grown sufficiency range. **High concentrations compared to *L. sativa* soil-grown sufficiency range.
(XLSX)

# Acknowledgments

This research was made possible due to strategic collaborations fostered by Claudia Ludwig.

# Author Contributions

**Conceptualization:** Jessica A. Day, Anne E. Otwell, Sean M. Gibbons, Nitin S. Baliga.

**Formal analysis:** Jessica A. Day, Christian Diener, Brad Bebout, Michael D. Lee, Sean M. Gibbons.

**Funding acquisition:** Brad Bebout, Nitin S. Baliga.

**Investigation:** Jessica A. Day, Christian Diener, Anne E. Otwell, Kourtney E. Tams, Brad Bebout, Angela M. Detweiler, Michael D. Lee, Madeline T. Scott, Wilson Ta, Monica Ha, Shienna A. Carreon, Kenny Tong, Abdirizak A. Ali, Sean M. Gibbons, Nitin S. Baliga.

**Methodology:** Jessica A. Day, Christian Diener, Anne E. Otwell, Brad Bebout, Angela M. Detweiler, Michael D. Lee, Sean M. Gibbons, Nitin S. Baliga.

**Project administration:** Jessica A. Day, Nitin S. Baliga.

**Resources:** Brad Bebout, Angela M. Detweiler, Nitin S. Baliga.

**Software:** Christian Diener, Michael D. Lee, Sean M. Gibbons.

**Supervision:** Jessica A. Day, Brad Bebout, Nitin S. Baliga.

**Visualization:** Christian Diener, Sean M. Gibbons.

**Writing – original draft:** Jessica A. Day, Christian Diener, Anne E. Otwell, Brad Bebout, Michael D. Lee, Sean M. Gibbons, Nitin S. Baliga.

**Writing – review & editing:** Jessica A. Day, Christian Diener, Anne E. Otwell, Kourtney E. Tams, Brad Bebout, Angela M. Detweiler, Michael D. Lee, Sean M. Gibbons, Nitin S. Baliga.

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
