## [Decision Letter · Decision Letter 0]

8 Dec 2020

PONE-D-20-27633

Inoculation strategy influences aquaponics productivity under nitrogen-limited conditions

PLOS ONE

Dear Dr.Sean Michael Gibbons：

Thank you for submitting your manuscript to PLOS ONE. After careful consideration, we feel that it has merit but does not fully meet PLOS ONE’s publication criteria as it currently stands. Therefore, we invite you to submit a revised version of the manuscript that addresses the points raised during the review process.

We look forward to receiving your revised manuscript.

Kind regards,

Zhili He, Ph.D.

Academic Editor

PLOS ONE

Journal Requirements:

Reviewers' comments:

Reviewer's Responses to Questions

**Comments to the Author**

1. Is the manuscript technically sound, and do the data support the conclusions?

Reviewer #1: Partly

Reviewer #2: Partly

2. Has the statistical analysis been performed appropriately and rigorously? 

Reviewer #1: No

Reviewer #2: Yes

3. Have the authors made all data underlying the findings in their manuscript fully available?

Reviewer #1: Yes

Reviewer #2: Yes

4. Is the manuscript presented in an intelligible fashion and written in standard English?

Reviewer #1: Yes

Reviewer #2: Yes

5. Review Comments to the Author

Reviewer #1: In this manuscript, the authors investigated how the inoculations of the different microbial consortium influence aquaponics productivity in the nitrogen-limited conditions. They compared the plant growth phenotypes, water chemistry, and microbiome composition in aquaponic systems inoculated with CIT and EIT over 61 days. By using amplicon sequencing analyses and association tests, they conclude that inoculation with EIT, rather than CIT， results in lower aquaponics productivity. While this discovery is interesting and might find a potential strategy to increase aquaponics productivity, this study seems to be a bit preliminary and would need substantial revisions before it can be published.

1. TITLE: Authors compared two different microbial consortium and their influences on aquaponics productivity, but this would not be considered as different inoculation strategies. I would suggest that authors should highlight the different inoculated microbial consortium rather than strategies in the title.

2. ABSTRACT：The background information in abstract is too much, authors should supplement more statements on data description and conclusions that were observed in this study.

3. INTRODUCTION: The introduction should be expanded to provide the reader with enough background information to understand why authors designed the whole experiment under the nitrogen-limited conditions. We would like to know more about the importance of nitrogen-limited conditions in the aquaponic systems.

4. RESUTLS AND DISCUSSION: Line 114, Figure 1d should be cited in the correct site of this manuscript; Line 121, here authors should add the association test data to illustrate the relations between low levels of nitrogen cycling with the death of a fish; in figure 2b, how the authors make sure that the nitrifying candidates only contains Nitrobacter, Nitrospmonas and Nitrospira. Line 148, It is a bit confusing for me that samples were clustered by time points in figure 3b, because we cannot find any different signals to stand for samples collected from different time points; Line 164-186, authors used association tests to find taxa that may be associated with those biochemical parameters, and set this results as one of the most important findings. However, figure 3e tells us that some of the associate tests may be just based on 2 samples with 3 replicates. If so, I suppose that the associate tests are not reasonable because of the limited number of samples and the links seem a bit thin. I would suggest that authors should explain it and add more detailed description in the methods part.

5. MATERIALS AND METHODS: Line 218-233, although authors carefully described IACUC protocol, we want to know how the biochemical parameters（e.g. water chemistry and plant growth metrics）were measured. The relative information is missing in the current version of manuscript.

Reviewer #2: Major Comments:

This study investigates how inoculation strategy influences aquaponics productivity. They show that inoculation from a pre-existing system results in lower crop yields than a commercial inoculum. The results are interesting, but I have several concerns:

1. The sample size is too small to support the conclusion of this study.

2. The significantly decreased cop yield is not necessarily due to bacteria. Other factors, such as fungi, viruses, or protists, can also have significant impacts. In other words, the authors should better characterize the nature of the inoculum, especially for the pre-existing system.

Minor comments:

Line 97 -112: most of the content should move to Materials and Methods;

Line 115 – 116: not clear how this was done. Did the author pool all physicochemical

properties together in a single test?

6. PLOS authors have the option to publish the peer review history of their article (what does this mean?). If published, this will include your full peer review and any attached files.

Reviewer #1: No

Reviewer #2: No

---

## [Author Response · Author response to Decision Letter 0]

18 Jan 2021

5. Review Comments to the Author

Reviewer #1: In this manuscript, the authors investigated how the inoculations of the different microbial consortium influence aquaponics productivity in the nitrogen-limited conditions. They compared the plant growth phenotypes, water chemistry, and microbiome composition in aquaponic systems inoculated with CIT and EIT over 61 days. By using amplicon sequencing analyses and association tests, they conclude that inoculation with EIT, rather than CIT， results in lower aquaponics productivity. While this discovery is interesting and might find a potential strategy to increase aquaponics productivity, this study seems to be a bit preliminary and would need substantial revisions before it can be published.

We thank the reviewer for their constructive comments, which have strengthened the quality and clarity of our work. Please see below for a point-by-point response to each reviewer concern.

TITLE: 

REVIEWER 1 CRITIQUE 1. Authors compared two different microbial consortium and their influences on aquaponics productivity, but this would not be considered as different inoculation strategies. I would suggest that authors should highlight the different inoculated microbial consortium rather than strategies in the title.

RESPONSE 1. We have changed the title of the manuscript to ‘Lettuce (Lactuca sativa) productivity influenced by microbial inocula under nitrogen-limited conditions in aquaponics’ based on this comment. 

ABSTRACT: 

REVIEWER 1 CRITIQUE 2. The background information in abstract is too much, authors should supplement more statements on data description and conclusions that were observed in this study.

RESPONSE 2. We have revised the abstract based on this comment. We removed a sentence that was focused on background information, and we made other modifications as indicated in red text in the revised manuscript. Our major conclusions are all highlighted in the abstract. 

INTRODUCTION: 

REVIEWER 1 CRITIQUE 3. The introduction should be expanded to provide the reader with enough background information to understand why authors designed the whole experiment under the nitrogen-limited conditions. We would like to know more about the importance of nitrogen-limited conditions in the aquaponic systems.

RESPONSE 3. Our aquaponic systems were nitrogen-limited due to our Institutional Animal Care and Use Committee (IACUC) protocol, which prioritizes animal welfare (i.e., by limiting the number of animals per tank). We have now stated this explicitly in the introduction (please see lines 83-86). It is also described in the Results/Discussion section (lines 102-103) and in the Materials and Methods (lines 233-237). While this is a limitation of our study, it is likely a common feature of experiments focused on aquaponic systems designed for research purposes (due to similar animal welfare protocols). We now discuss this limitation/qualification in our concluding paragraph (lines 105-200).

RESULTS AND DISCUSSION: 

REVIEWER 1 CRITIQUE 4. Line 114, Figure 1d should be cited in the correct site of this manuscript

RESPONSE 4. Thank you for pointing this out. Figure 1d is now referenced in line 96. 

REVIEWER 1 CRITIQUE 5. Line 121, here authors should add the association test data to illustrate the relations between low levels of nitrogen cycling with the death of a fish

RESPONSE 5. We have now added the particular correlation coefficient in the text at lines 157-159. However, it should be noted that this particular correlation test did not pass our significance threshold of p < 0.05 (p=0.1), so it remains unclear whether the number of dead fish had an impact on the nitrate levels in the system.

REVIEWER 1 CRITIQUE 6. In figure 2b, how the authors make sure that the nitrifying candidates only contains Nitrobacter, Nitrosomonas and Nitrospira. 

RESPONSE 6. The particular taxa were chosen because the provider of the commercial inoculum claimed they were present in the starter culture. Those taxa are also often the major nitrifiers in aquaponics systems (Wongkiew et al. 2018). We agree with the reviewer that there may be other taxa that act as nitrifiers in the system. For instance, archaea were not quantified in our study, and therefore species of ammonia-oxidizing archaea could have been missed (this is now added as a qualification in our concluding paragraph, lines 187-188). However, the generally low levels of nitrification support the hypothesis that all systems were nitrogen-limited and showed a paucity of nitrifying taxa. 

Wongkiew S, Park M-R, Chandran K, Khanal SK. Aquaponic Systems for Sustainable Resource Recovery: Linking Nitrogen Transformations to Microbial Communities. Environ Sci Technol. 2018;52: 12728–12739.

REVIEWER 1 CRITIQUE 7. Line 148, It is a bit confusing for me that samples were clustered by time points in figure 3b, because we cannot find any different signals to stand for samples collected from different time points

RESPONSE 7. We agree with the reviewer. Figure 3B does not indicate any particular clustering by timepoint and we thus removed this erroneous statement from the text (Sentence removed from line 131 in revised manuscript). 

REVIEWER 1 CRITIQUE 8. Line 164-186, authors used association tests to find taxa that may be associated with those biochemical parameters, and set this results as one of the most important findings. However, figure 3e tells us that some of the associate tests may be just based on 2 samples with 3 replicates. If so, I suppose that the associate tests are not reasonable because of the limited number of samples and the links seem a bit thin. I would suggest that authors should explain it and add more detailed description in the methods part.

RESPONSE 8. While we agree that the study includes relatively few samples, we respectfully disagree that we only studied “2 samples with 3 replicates”. Association tests were run for a total of 6 independent aquaponics systems divided into two different inoculation strategies. We think some of the confusion may have come from overlapping points in Figure 3E. Thus, every test for a specific taxon included 6 samples (3 per group). We have now added some horizontal dodging and strokes to make all samples visible in this plot. Please see revised Figure 3E below.

Additionally, we took care to choose a strategy that would yield the highest possible statistical power along with a conservative control of the false discovery rate. Here we used DESeq2 which estimates variances of individual taxa by a bayesian approach that uses pooled information from all the available taxa (hundreds in our case) in order to stabilize the per-taxa variance estimate. As shown by Love et. al. (https://doi.org/10.1186/s13059-014-0550-8, Fig. 6) this will yield sufficient control of the false discovery rate even for a setting with 6 samples (such as ours) as long as the fold change is >2 (absolute log fold-change >1). This comes at the expense of lower sensitivity, meaning it is likely we missed some significant taxa, but the ones that were identified are likely correct. In the end, statistical power is a function of sample size and effect size and the reported effect sizes are very large. For instance, Pseudomonas switches from being a low-abundance taxon (<6% abundance on average) in the CIT samples to being the dominant taxon in the roots of EIT systems (>70% on average). 

MATERIALS AND METHODS: 

REVIEWER 1 CRITIQUE 9. Line 218-233, although authors carefully described IACUC protocol, we want to know how the biochemical parameters（e.g. water chemistry and plant growth metrics）were measured. The relative information is missing in the current version of manuscript.

RESPONSE 9. We have added a new section in the Materials and Methods (‘Lettuce phenotypes’) in order to address this comment. Please see lines 269-274. Water chemistry measurements are described in the section of the Materials and Methods titled, ‘Water chemistry maintenance’ (lines 257-267). 

Reviewer #2: Major Comments:

This study investigates how inoculation strategy influences aquaponics productivity. They show that inoculation from a pre-existing system results in lower crop yields than a commercial inoculum. The results are interesting, but I have several concerns:

We thank the reviewer for their constructive comments, which have strengthened the quality and clarity of our work. Please see below for a point-by-point response to each reviewer concern.

REVIEWER 2 CRITIQUE 10. The sample size is too small to support the conclusion of this study.

RESPONSE 10. Please see Response 8, which also discusses this critique. We carefully chose a strategy that would ensure sufficient control of the false discovery rate even when comparing a total of 6 samples as in our case. This was likely at the expense of some sensitivity. However, the observed fold-changes are much larger than what is required by DESeq2 for decent statistical power, and we therefore have confidence that the observed differences that we discuss are indeed present. We cannot, however, rule out that certain taxa were not detected due to the low sample number.

REVIEWER 2 CRITIQUE 11. The significantly decreased cop yield is not necessarily due to bacteria. Other factors, such as fungi, viruses, or protists, can also have significant impacts. In other words, the authors should better characterize the nature of the inoculum, especially for the pre-existing system.

Line 46, 189

RESPONSE 11. It is true that the reduced plant growth that we observed is not necessarily directly related to bacteria. We make certain to discuss relationships between lettuce productivity and the microbial community as correlations, as opposed to causations. We have now emphasized the point that fungi, viruses, or protists could be playing significant (and unmeasured) roles (please see lines 179-180 and lines 185-188). 

Minor comments:

REVIEWER 2 CRITIQUE 12. Line 97 -112: most of the content should move to Materials and Methods;

RESPONSE 12. Thank you for this helpful comment. We have moved this content out of the Results and Discussion and into the Materials and Methods section. 

REVIEWER 2 CRITIQUE 13. Line 115 – 116: not clear how this was done. Did the author pool all physicochemical properties together in a single test?

RESPONSE 13. This was done by testing each physiochemical property separately for a significant difference between CIT and EIT systems. Significance was assessed using t-tests. The text has been modified (line 98) to make it clear that each physiochemical property was assessed individually, and we have also added the individual p-values for each panel in Supplementary Figure 1. Please see this revised figure and caption below.

Figure S1. Water chemistry, environmental parameters, and system inputs measured throughout the study period. Lines denote LOESS smoothed curves for each inoculum and bands denote 95% confidence intervals of the regression. Indicated p-values were obtained from individual t-tests of CIT vs EIT systems.

---

## [Decision Letter · Decision Letter 1]

9 Feb 2021

Lettuce (Lactuca sativa) productivity influenced by microbial inocula under nitrogen-limited conditions in aquaponics

PONE-D-20-27633R1

Dear Dr. Gibbons,

We’re pleased to inform you that your manuscript has been judged scientifically suitable for publication and will be formally accepted for publication once it meets all outstanding technical requirements.

Kind regards,

Zhili He, Ph.D.

Academic Editor

PLOS ONE

Additional Editor Comments (optional):

Reviewers' comments:

Reviewer's Responses to Questions

**Comments to the Author**

1. If the authors have adequately addressed your comments raised in a previous round of review and you feel that this manuscript is now acceptable for publication, you may indicate that here to bypass the “Comments to the Author” section, enter your conflict of interest statement in the “Confidential to Editor” section, and submit your "Accept" recommendation.

Reviewer #1: All comments have been addressed

Reviewer #2: All comments have been addressed

2. Is the manuscript technically sound, and do the data support the conclusions?

Reviewer #1: Yes

Reviewer #2: Partly

3. Has the statistical analysis been performed appropriately and rigorously? 

Reviewer #1: Yes

Reviewer #2: Yes

4. Have the authors made all data underlying the findings in their manuscript fully available?

Reviewer #1: Yes

Reviewer #2: Yes

5. Is the manuscript presented in an intelligible fashion and written in standard English?

Reviewer #1: Yes

Reviewer #2: Yes

6. Review Comments to the Author

Reviewer #1: (No Response)

Reviewer #2: The authors have made some changes that address my comments. I am still skeptical about the sample size. The rest looks fine.

7. PLOS authors have the option to publish the peer review history of their article (what does this mean?). If published, this will include your full peer review and any attached files.

Reviewer #1: No

Reviewer #2: No

---

## [Editor Report · Acceptance letter]

12 Feb 2021

PONE-D-20-27633R1 

Lettuce (*Lactuca sativa*) productivity influenced by microbial inocula under nitrogen-limited conditions in aquaponics 

Dear Dr. Gibbons:

I'm pleased to inform you that your manuscript has been deemed suitable for publication in PLOS ONE. Congratulations! Your manuscript is now with our production department. 

Kind regards, 

on behalf of

Dr. Zhili He 

Academic Editor

PLOS ONE